# Indirect Negative Effect of Mutant Ataxin-1 on Short- and Long-Term Synaptic Plasticity in Mouse Models of Spinocerebellar Ataxia Type 1

**DOI:** 10.3390/cells11142247

**Published:** 2022-07-20

**Authors:** Anton N. Shuvaev, Olga S. Belozor, Oleg I. Mozhei, Andrey N. Shuvaev, Yana V. Fritsler, Elena D. Khilazheva, Angelina I. Mosyagina, Hirokazu Hirai, Anja G. Teschemacher, Sergey Kasparov

**Affiliations:** 1Research Institute of Molecular Medicine and Pathobiochemistry, Krasnoyarsk State Medical University, 660022 Krasnoyarsk, Russia; olsbelor@gmail.com (O.S.B.); elena.hilazheva@mail.ru (E.D.K.); angelina.mosiagina@gmail.com (A.I.M.); 2Institute of Living Systems, Immanuel Kant Baltic Federal University, 236041 Kaliningrad, Russia; vulpecula999@gmail.com (O.I.M.); sergey.kasparov@bristol.ac.uk (S.K.); 3Institute of Fundamental Biology and Biotechnology, Siberian Federal University, 660041 Krasnoyarsk, Russia; andrey.n.shuvaev@gmail.com (A.N.S.); fri.yana@mail.ru (Y.V.F.); 4Department of Neurophysiology and Neural Repair, Gunma University Graduate School of Medicine, Maebashi 371-8511, Japan; hirai@gunma-u.ac.jp; 5Department of Physiology, Pharmacology, and Neuroscience, University of Bristol, Bristol BS8 1TH, UK; anja.teschemacher@bristol.ac.uk

**Keywords:** spinocerebellar ataxia type 1, PPF, PPD, DSE, LTD, synaptic plasticity, Purkinje cells, Bergmann glia

## Abstract

Spinocerebellar ataxia type 1 (SCA1) is an intractable progressive neurodegenerative disease that leads to a range of movement and motor defects and is eventually lethal. Purkinje cells (PC) are typically the first to show signs of degeneration. SCA1 is caused by an expansion of the polyglutamine tract in the *ATXN1* gene and the subsequent buildup of mutant Ataxin-1 protein. In addition to its toxicity, mutant Ataxin-1 protein interferes with gene expression and signal transduction in cells. Recently, it is evident that *ATXN1* is not only expressed in neurons but also in glia, however, it is unclear the extent to which either contributes to the overall pathology of SCA1. There are various ways to model SCA1 in mice. Here, functional deficits at cerebellar synapses were investigated in two mouse models of SCA1 in which mutant ATXN1 is either nonspecifically expressed in all cell types of the cerebellum (SCA1 knock-in (KI)), or specifically in Bergmann glia with lentiviral vectors expressing mutant ATXN1 under the control of the astrocyte-specific GFAP promoter. We report impairment of motor performance in both SCA1 models. In both cases, prominent signs of astrocytosis were found using immunohistochemistry. Electrophysiological experiments revealed alteration of presynaptic plasticity at synapses between parallel fibers and PCs, and climbing fibers and PCs in SCA1 KI mice, which is not observed in animals expressing mutant ATXN1 solely in Bergmann glia. In contrast, short- and long-term synaptic plasticity was affected in both SCA1 KI mice and glia-targeted SCA1 mice. Thus, non-neuronal mechanisms may underlie some aspects of SCA1 pathology in the cerebellum. By combining the outcomes of our current work with our previous data from the B05 SCA1 model, we further our understanding of the mechanisms of SCA1.

## 1. Introduction

Spinocerebellar ataxia type 1 (SCA1) is an inherited neurodegenerative disease caused by mutations in the ATXN-1 gene which encode an abnormally expanded polyglutamine (polyQ) tract in the protein Ataxin-1 [1,2]. SCA1 is characterized by progressive cerebellar ataxia, followed by bulbar paralysis and death [3]. Ataxin-1 is a transcriptional coactivator and operates in a complex with other proteins, including retinoid-related orphan receptor α (RORα), which is abundantly expressed in cerebellar Purkinje cells (PCs). The mutant Ataxin-1 fails to interact with RORα which leads to the downregulation of expression of a number of downstream genes including important players of the Ca^2+^ signaling cascade, such as mGluR1 and IP3R [4].

Purkinje cells in the cerebellar cortex are principal neurons fundamental for the control of balance and coordination. These cells are typically first affected in SCA1. Historically, various approaches have been developed to model SCA1 pathology in animal models (Appendix A). One such model is the SCA1 B05 mouse which expresses 82-polyQ-Ataxin-1 selectively in Purkinje cells (Appendix A). For this reason, SCA1 B05 mice are a good model to investigate molecular pathology specifically at the level of Purkinje cells and the impact of this pathology on the ataxic phenotype [5]. Our previous study showed dysfunction in cerebellar parallel fiber (PF)-PCs synapses in B05 mice from postnatal week five. At 12 weeks of age, SCA1 B05 mice exhibited a significant impairment of mGluR1 signaling and a reduction in synaptic plasticity [6].

While initially the attention was focused on neurons, during the past decade it has been shown that astrocytes are also involved in SCA1 pathology [7,8]. Previously a new model was developed which utilizes lentiviral vectors (LVV) expressing mutant Ataxin-1 selectively in astrocytes, which are known as Bergman glia (BG) in the cerebellum (Appendix A). The expression of mutant ATXN1 made glia reactive, leading to PCs degeneration and altered synaptic transmission and plasticity [9]. 

However, in SCA1 patients, polyQ-Ataxin-1 is expressed in both neurons and glia. This is replicated in the SCA1 KI mouse model where 154-polyQ-Ataxin-1 is knocked directly into the endogenous locus of this gene. As a result, the mutant appears in all cell types of CNS (Appendix A). This may be seen as a more realistic representation of SCA1 pathology [10,11]. 

In the present study, we compare polyQ-Ataxin-1-evoked changes in cerebellar histology and synaptic plasticity between the SCA1 KI mouse and glia-targeted LVV-SCA1 models [9] in order to gain a better understanding of the contribution of neuronal versus glial compartments to the SCA1-associated neurodegeneration. This allows us to compare these findings with our previously published data obtained from the B05 mouse model [6].

## 2. Materials and Methods

All procedures for the care and handling of animals were in accordance with the European and Japanese Acts on the Welfare and Management of Animals and the Guidelines for the Proper Conduct of Animal Experiments issued by the Science Council of Japan. The experimental protocol was approved by Gunma University Animal Care and Experimentation Committee (07-015 and 04-44) and the Krasnoyarsk State Medical University and Russian public standard (33215–2014) regulations, and were approved by the local ethical committee. All efforts were made to minimize suffering and reduce the number of animals used in this study. 12-week-old CD-1 IGS wild type (WT) mice (Charles River Laboratories) (15 and 14 animals in LVV Ataxin 1[Q2] and Ataxin1[Q85] injected groups, respectively) and 12-week-old C57BL/6 SCA1 KI mice with 154 CAG repeats in the endogenous locus of ATXN1 gene [10] and their WT littermates were used in this study (15 animals in each group). All groups of mice contained males and females at approximately even numbers, no differences were noted and data were pooled. Animals were kept on a 12-h light/dark cycle with free access to food and water.

### 2.1. LVV Production and Amplification

In order to target mutant ATXN1 to astroglia specifically, we used a lentiviral vector where the transgene was placed under the control of astrocyte-specific enhanced GFAP promoter [12]. Sequences of non-pathogenic ATXN1[Q2] (encoding human ataxin-1 with 2 glutamine repeats) or pathogenic ATXN1[Q85] (with 85 uninterrupted glutamine repeats) were fused in frame with the sequence encoding the FLAG tag at their 5′ends. Next, Flag-ATXN1[Q2] and Flag-ATXN1[Q85] constructs were transferred into the lentiviral shuttle vector pTYF, under the control of the enhanced GFAP promoter. The detailed procedure for viral vector production was described previously [13]. Titers of LVV-GFAP-Flag-ATXN1[Q2] LVV and LVV-GFAP-Flag-ATXN1[Q85] were 7 × 10^9^ transducing units (TU)/mL. LVV were stored at −80 °C and used within 6 months.

### 2.2. LVV Injections

Three-week-old (P21) WT mice were anesthetized by Zoletil (50 mg/kg i.p.; Virbac, Carros, France). The level of anesthesia was controlled by monitoring the lack of withdrawal reflex every few minutes and the anesthetic was supplemented as required. Mice were kept warm by a heated pad during surgical interventions. 3μL of LVV or PBS were slowly injected into the cortex of the cerebellar vermis (lobule VI) using a 10μL Hamilton syringe. Stereotaxic coordinates relative to bregma were: AP: −2.5 mm, ML: 0 mm, DV: 2 mm. Mice were used for further experiments 9 weeks after the injection when expression of transgenic Ataxin-1 was prominent. Immunostaining against Flag was used to visualize the expression of LVV constructs (Appendix A).

### 2.3. Rotarod Test

The motor behavior was analyzed by a rotarod test for two consecutive days. The accelerating speed protocol [3 min acceleration from 4 to 40 revolutions per minute (rpm)] comprised 4 trials, with a 30 min interval between trials. A slow-starting, accelerating mode was chosen to account for animals with impaired coordination such as SCA1 KI mice, which may fall off immediately if placed on a quickly turning rod. [14]. The maximum duration of each trial was set to 300 s. Time spent on the rod was recorded. If a mouse remained on the rod for the entire trial, a 300 s latency time was recorded. Time-averaged across all the trials per each day was used for the statistical analysis.

### 2.4. Acute Slice Preparation

Cerebellar slices (200 μm) were prepared, and whole-cell recordings were conducted as described previously [15]. Briefly, mice were deeply anesthetized by Zoletil (50 mg/kg) intraperitoneally (Virbac, Carros, France) and killed by decapitation. The whole brain was quickly dissected out and immersed for several minutes in ice-cold Ringer’s solution containing (in mM): 234 sucrose, 26 NaHCO_3_, 2.5 KCl, 1.25 NaH_2_PO_4_, 11 glucose, 10 MgSO_4_, and 0.5 CaCl_2_ 0.5; pH 7.4, continuously oxygenated by 95% O_2_ and 5% CO_2_. Parasagittal slices of cerebellar vermis were made using a microslicer (Thermo Scientific, Waltham, MA USA; Microtom CU65). The slices were maintained in an extracellular solution containing (in mM): 125 NaCl, 2.5 KCl, 2 CaCl_2_, 1 MgCl_2_, 1.25 NaH_2_PO_4_, 26 NaHCO_3_, 10 D-glucose, and 0.05–0.1 picrotoxin. This solution was oxygenated continuously with a mixture of 95% O_2_ and 5% CO_2_ at room temperature for 1h before starting the electrophysiological experiments.

### 2.5. Electrophysiology

For current-clamp whole-cell recordings from PCs we used K-gluconate-based intracellular solution containing (in mM): 130 K-gluconate, 4 KCl, 20 HEPES, 1 MgCl_2_, 4 MgATP, 1 NaGTP, 0.4 EGTA (pH 7.3 adjusted with KOH). For voltage-clamp whole-cell recordings from PCs we used intracellular solution containing (in mM): 140 Cs-gluconate, 8 KCl, 10 HEPES, 1 MgCl2, 2 MgATP, 0.4 NaGTP, 0.2 EGTA (pH 7.3 adjusted with CsOH). The passive electrical properties of the PCs were estimated using averaged traces of ~20 current responses (acquisition; low-pass filtered at 10 kHz and sampled at 50 kHz) and evoked by hyperpolarising voltage pulses (from −70 to −80 mV, 200 ms duration). Liquid junction potentials were not corrected in this study. Analysis of electrophysiological data was performed using pClamp10 (Molecular Devices), Patchmaster software (HEKA), and Clampfit 10.5 (Axon Instruments).

PCs were voltage-clamped at −70 mV to record parallel fiber (PF) EPSCs and at +10 mV to record climbing fiber (CF) EPSCs. We placed the stimulation electrode in ML to evoke PF EPSCs and into GL to evoke CF EPSCs. Selective stimulation of CFs and PFs was confirmed by paired-pulse depression and paired-pulse facilitation of EPSC amplitudes (at a 50-ms interstimulus interval), respectively. For the recordings of mGluR1-mediated slow EPSCs, the strength of the electrical stimulation was adjusted to produce AMPA receptor-mediated fast EPSCs with amplitudes of ~300 pA.

For analysis of LTD, PF EPSCs were monitored every 10 s. To induce LTD, we applied 30 single PF stimuli paired with single 200 ms depolarizing pulses (−70 to +20 mV) repeated at 1 Hz. Averaged amplitudes of PF EPSCs over 1 min were normalized to their baseline values, which were the averages of six traces just before the LTD induction.

To examine short-term synaptic depression (DSE), PF EPSCs were recorded every 3 s. After monitoring basal PF EPSCs for 1 min, a single standard depolarizing step (5 s from −70 to 0 mV) was applied to evoke DSE. Amplitudes of subsequent PF EPSCs were normalized to the mean basal value of 12 control PF EPSCs recorded before the induction.

### 2.6. Immunohistochemistry

For immunohistochemistry, mice were perfused transcardially with a fixative containing 4% paraformaldehyde in 0.1 M phosphate buffer after being anesthetized by Zoletil (50 mg/kg) intraperitoneally (Virbac, Carros, France). The whole brain was removed and postfixed in the same fixative for 5–6 h or overnight. The cerebellar vermis was cut into 50-µm sagittal sections. The sections were treated with rabbit monoclonal anti-calbindin D-28 k (1:500, Cloud Clone Corp., China), chicken polyclonal anti-GFAP (1:1000, Abcam, UK), rabbit polyclonal anti-Flag (1:500, Cloud Clone Corp., China) or anti-S100β (1:500, Cloud Clone Corp., China) antibodies and then visualized with Alexa Fluor 594-conjugated donkey anti-mouse t IgG (1:1000, Life Technologies), Alexa Fluor 488-conjugated donkey anti-rabbit IgG (1:1000, Life Technologies) or Alexa Fluor 647-conjugated goat anti-chicken IgG (1:1000, Life Technologies). The antibodies were dissolved in a PBS solution containing 2% (*v*/*v*) normal donkey serum, 0.1% (*v*/*v*) Triton X-100, and 0.05% NaN_3_. Confocal fluorescence images of the cerebellar slices were obtained from the corresponding region of the cerebellum for comparison. Fluorescent images were obtained using FV10i Confocal Microscope (Olympus, Japan). Images were recorded as Z-stacks using ×10 objective and 1024 × 1024 resolution.

### 2.7. Data Analysis

Pooled data are expressed as M ± S.E.M. mean values with 95% confidence interval. Statistical analyses of differences between the individual groups were performed using unpaired *t*-test. Statistical analyses of differences between multiple groups were performed using the one-way or two-way ANOVA followed by Tukey’s post hoc test. The Bonferroni adjustment was made for derived *p*-values using the R package. Differences were considered significant at adjusted *p* < 0.05.

## 3. Results

### 3.1. Expression of Mutant ATXN1 Disrupts Motor Coordination 

Expression of ATXN1[Q85] in BG significantly affected the rotarod performance of mice from day one. On day two, as typical for normal mice, and as seen in both controls used in these experiments, ATXN1[Q85] mice showed an improvement in performance, the average time spent on the rod was 129.9 ± 12.4 s (*n* = 11) compared to 176.6 ± 9.1 s (*n* = 13) in animals which were transduced with ATXN1[Q2] (*p* = 0.007; Figure 1A). Transgenic SCA1 KI mice showed a comparably compromised baseline performance on day one but completely failed to increase their motor performance on day two (*p* = 0.14). On day two, KI mice remained on the rod for 79.6 ± 4.8 s (*n* = 11), while their WT counterparts achieved 231.1 ± 10.6 s (*n* = 11, *p* = 3.6 × 10^−9^, Figure 1B).

### 3.2. Expression of Mutant ATXN1 Selectively in BG and Non-Selectively as in SCA1 KI Model Affects Morphology and Electrophysiological Properties of PCs

Calbindin staining is commonly used to visualize PCs and assess their morphology (Figure 2A,B). LVV-mediated expression of ATXN1[Q85] selectively in BG reduced the thickness of the molecular layer (ML) from 178.4 ± 3.2 µm (*n* = 23 from four animals) to 161.7 ± 2.2 µm (*n* = 23 from four animals, *p* = 0.0001; Figure 2C) due to the collapse of PCs dendrites. 

In SCA1 KI mice the ML thickness was reduced to 144.0 ± 2.2 μm (*n* = 14 from three animals) in comparison to WT controls (174.8 ± 4.5 μm; *n* = 11 from three animals; *p* = 2.1 × 10^−5^; Figure 2D). 

These morphological differences corresponded to the drop in PC membrane capacitance (Cm) measured in cerebellar slices by patch-clamp, which was significantly reduced in both LVV-GFAP-Flag-ATXN1[Q85] and SCA1 KI mice (*p* = 0.002 and *p* = 0 0012, correspondingly, Figure 2D,F and Table 1). 

These changes indicate severe impairments in PC morphology in both models of SCA1 and are consistent with our previous observations from another SCA1 model, the B05 mouse [6].

Other passive membrane properties such as membrane and access resistance did not differ significantly between genotypes (Table 1).

### 3.3. Expression of Mutant ATXN1 Effects Short Term Synaptic Plasticity

In the cerebellum, granule cells establish PF-PC synapses, and neurons from inferior olives establish CF-PC synapses with PCs. In these synapses glutamate is released, which activates AMPA receptors on PCs, leading to fast excitatory postsynaptic currents (EPSCs) [16]. We found that amplitudes and kinetics of EPSCs in either pathway were not affected in our SCA1 models (Table 2).

Glutamate release is triggered by Ca^2+^ entry via voltage-gated Ca^2+^ channels (VGCCs) on presynaptic terminals. Two stimuli applied to the PF within 50 ms lead to an accumulation of free presynaptic Ca^2+^ in granule cell axons, resulting in enhanced glutamate release in response to the second stimulus [17]. This is known as paired pulse facilitation (PPF) and is thought to involve processes selectively within presynaptic elements of the axons of the granule cells [18]. In animals expressing ATXN1[Q85] selectively in BG, the PPF ratio in PF-PC synapses was not different from the control group, *p* = 0.78 (Figure 3A). In contrast, in SCA1 KI mice where the mutant Ataxin-1 is also expressed in granule cells, the PPF ratio decreased from 1.6 ± 0.1 (*n* = 11 from 3 cells) to 1.3 ± 0.1 (*n* = 11 from 3 cells, *p* = 0.027, Figure 3B). 

On the other hand, CF synapses, formed by axons of inferior olivary nucleus neurons with PC, typically display a reduction of the second response, known as paired pulse depression (PPD). PPD is associated with depletion of presynaptic vesicular glutamate which lasts tens of milliseconds [19]. In animals expressing ATXN1[Q85] selectively in BG, the PPD ratio in CF-PC synapses was not different from the control group (*p* = 0.46; Figure 4A). However, in SCA1 KI mice where the mutant Ataxin-1 is expressed also in neurons, including those of the inferior olive, PPD was enhanced, with the ratio 0.64 ± 0.02 (*n* = 11 from 3 animals) in control versus 0.56 ± 0.03 (*n* = 10 from 3 animals, *p* = 0.043) in SCA1 KI animals (Figure 4B).

These data demonstrate that the ubiquitous expression of mutant Ataxin-1 in SCA1 KI mice affects excitatory transmission between granule cells, inferior olive neurons, and the PC. This aspect of SCA1 pathology is only recapitulated by SCA1 KI mice but not in other models, including B05 [6]. 

### 3.4. Mutant Ataxin-1 in BG Compromises Postsynaptic Short-Term Plasticity in PCs

Depolarization of PC membrane causes Ca^2+^ influx through VGCC and this can trigger the release of endocannabinoids, which signal retrogradely to inhibit the release of glutamate from the presynaptic site. This plasticity is known as depolarization-induced depression of excitation (DSE). 

Astrocytes have been implicated in endocannabinoid signaling [20]. Specifically, astrocytes express CB1 receptors which, when activated by endocannabinoids, trigger an increase in intracellular Ca^2+^ levels and release of gliotransmitters, including glutamate and possibly D-serine and others, which contribute to the effects seen at the postsynaptic level [21,22]. Therefore it was hypothesized that mutant Ataxin-1 expressed in BG (through LVV-GFAP-Flag-ATXN1[Q85]) or ubiquitously (SCA1 KI mouse) would change DSE compared to the previously studied B05 mice where mutant Ataxin-1 is specifically localized in PC.

The reactive phenotype of BG was revealed using anti-S100β staining. In both our SCA1 models, the expression of S100β was increased (Figure 5A,B). This is consistent with previous research, demonstrating prominent BG reactivity in SCA1 KI mice [8,23]. Following the DSE stimulus protocol, EPSCs dropped to a similar level (47.5% and 43.3% in control mice used for both models; Figure 5C,D). LVV-mediated expression of ATXN1[Q85] selectively in BG, did not significantly reduce the peak DSE values; Figure 5C). However, the recovery from the depression occurred much faster when BG was expressing ATXN1[Q85] where the EPSC returned to pre-control values after 15 s in contrast to the controls where it lasted up to 65 s (*p* = 0.28 and *p* = 0.12, respectively, one way ANOVA). In contrast, in KI mice, DSE was markedly reduced and the PF EPSC amplitude dropped to only 84.6 ± 4.2% (*n* = 10 from three mice of control) compared to their WT counterparts (decrease to 42.3 ± 5.9%; *n* = 10 from four mice). After 15 s EPSC were already not statistically different from the pre-stimulation level (*p* = 0.11, measured by one-way ANOVA). 

Since the drop in the peak DSE in the SCA1 KI model was greater than in the case of selective targeting of the mutant Ataxin-1 to BG (bar chart inserts in Figure 5C,D), we further investigated the mechanism of this effect. One possibility is that in the KI model where the mutant is expressed in PC as well as BG, depolarization in response to current injection is weaker, resulting in reduced release of endocannabinoids which critically depends on the influx of Ca^2+^ into the neurons via voltage-gated Ca^2+^ channels (VGCC) [24].

In PCs of SCA1 KI mice, the frequency of action potentials in response to current injection was consistently lower in comparison to PCs of control littermates. To estimate the excitability of PCs while taking into account differences in membrane capacitance caused by the collapse of the dendritic tree, the injected currents were normalized to membrane capacitance (pA/pF). Firing frequency was plotted against this parameter. With smaller currents below 0.2 pA/pF, PCs from SCA1 KI mice did not generate any action potentials (Figure 6A). The input resistance in these two groups did not differ (data not shown), but the resting membrane potential of PCs in SCA1 KI mice was ~1.5 mV lower; *p* = 0.018; Figure 6B). 

These results point to a lower excitability of the diseased PCs which is likely to affect the opening of VGCC and could contribute to a weaker peak DSE. On the other hand, the maintenance of DSE clearly depends on the health of the BG.

### 3.5. Expression of Mutant ATXN1 Affects Long-Term Synaptic Plasticity in PF-PC Synapses

Long-term depression (LTD) at PF-PC synapses is one form of cerebellar synaptic plasticity [25]. To evoke LTD, we applied 30 single PF stimuli combined with single 200 ms depolarizing pulses (−70 to +20 mV) repeated at 1 Hz. This type of LTD requires postsynaptic mGluR1 signaling in PCs [26]. After 30 min of LTD induction, the normalized PF EPSC amplitudes were reduced to a significantly lesser degree in PCs from the animals transduced with LVV-GFAP-Flag-ATXN1[Q85] (80.1 ± 4.8%; *n* = 5 from three animals, compared to 56.2 ± 5.4% in ATXN1[Q2]-expressing animals; *n* = 5 from three animals; *p* = 0.026; Figure 7A). In SCA1 KI mice LTD was also reduced 30 min after induction (88.9 ± 6.2%; *n* = 6 from three mice; versus 67.3 ± 4.2%; *n* = 7 from three mice; *p* = 0.013; Figure 7B). Thus, PF-PC LTD is affected in both models to a similar degree.

## 4. Discussion

SCA1, like all similar pathologies, is a complex disease. The expression of the mutant Ataxin-1 is not limited to a specific subset of neurons even though it typically first manifests by signs caused by degeneration of the PCs. Modeling such a process is not an easy task and various approaches have been used previously by other groups and ourselves. These models use a variety of molecular strategies and therefore the patterns of expression of the mutant protein are different. In the case of the SCA1 KI mouse, exon 8 of the healthy allele was replaced with a mutant containing 154 CAG repeats [10]. The expectation was that, in that animal, the expression pattern of the mutant should largely follow the same rules as that of the endogenous ATXN1 gene. Hence, the expression is ubiquitous and not limited to PCs or any specific cell type in the brain. This mouse remains one of the best tools to study SCA1.

Another transgenic mouse model is the B05 mouse where mutant Ataxin-1 expression is targeted mainly to the PCs using a specific promoter, pcp2 [5]. This model is interesting because it allows us to investigate the consequences of mutant Ataxin-1 specifically on PCs. In our previous work, it was shown that B05 mice develop motor impairment [6]. Both, B05 and SCA1 KI, mice develop a prominent ataxic phenotype at 12 weeks of age [6] which was the age the current study focused on. 

In addition, the contribution of BG to the development of various aspects of SCA1 pathology was studied by expressing mutant Ataxin-1 specifically in these glial cells using our well-established LVV system with an enhanced astrocyte-specific GFAP promoter [12]. We previously confirmed the specificity of this system [9]. It was found that expression of ATXN1[Q85] selectively in BG causes signs of degeneration in the cerebellar cortex and strongly affects rotarod motor performance (Figure 1). 

Thus, by combining and comparing data from the three models, it is possible to begin to differentiate effects caused by the mutant Ataxin-1 in different compartments of the PCs tripartite synapses (Appendix A). 

By targeting mutant Ataxin-1 selectively to BG we demonstrate that an ataxic phenotype can develop without the direct impact of the mutant on the PCs (Figure 1). Moreover, polyQ-Ataxin-1 expressing BG not only turns reactive but also causes degeneration of PC morphology (Figure 2A). Similar signs of neurodegeneration developed also in the SCA1 KI mice (Figure 2B) and were previously reported by us in B05 mice [6]. Thinning of ML and decrease in PCs capacitance (Figure 2C–F) reflect a collapse of the dendritic tree of the PCs, which should affect both synaptic density and synaptic plasticity in excitatory glutamate synapses [27,28]. Since these morphological changes are very similar between all three models it may be concluded that PCs degeneration can develop without a direct impact of the mutant Ataxin-1 on PCs.

Mutant Ataxin-1 is likely to impair synaptic transmission in a variety of ways, for example by attenuating glutamate release from presynaptic terminals [29]. Here, two forms of plasticity in glutamatergic synapses, PPF and PPD were investigated. Interestingly, PPF and PPD ratios were not affected in B05 mice and mice which express ATXN1[Q85] selectively in BG (Figure 3A and Figure 4A, and [6]). However, SCA1 KI mice that express mutant Ataxin-1 ubiquitously, including in PF (axons of granular cells) and CF (axons of inferior olivary nucleus neurons), exhibited prominent changes in PPF and PPD ratios (Figure 3B and Figure 4B). PPF was reduced in KI mice while PPD seemed to be stronger than in control mice, possibly due to the reduced stores of vesicular glutamate in CF-PC synapses. It appears that these changes result mainly from the dysregulation of glutamate release in the afferent neurons, rather than from glutamate handling by the BG. Synaptic plasticity at the level of PCs is essential for motor learning, and such dysregulation can be expected to cause ataxia.

DSE is another interesting manifestation of plasticity in PCs-PF synapses. DSE is thought to be mainly dependent on the retrograde action of endocannabinoids, released after depolarization of the PCs [30]. Our group reported earlier that DSE in B05 mice was intact [6]. In the current study, we found that selective expression of the mutant Ataxin-1 in glia affects the duration of DSE more than its peak amplitude. In contrast, non-cell selective expression in SCA1 KI mice affected the peak DSE as well as the speed of recovery (Figure 5). In our previous work, injection of glial S100β protein into the cerebellar cortex led to DSE impairment [31]. There is a remarkable similarity in the outcomes of these two models which mainly mimic glia involvement (compare Figure 7A in [28] and Figure 5C in this paper). In both cases, the peak DSE values were hardly affected while the maintenance of PF EPSC inhibition was greatly reduced. It appears that DSE may be compromised in SCA1 due to the pathological processes within both, PCs and local astrocytes, the BG. We believe that DSE induction could be reduced due to the lower excitability of PCs membrane (Figure 6). There are several reports that point to hyperpolarization of PCs in SCA models. Specifically, the firing rate was altered in SCA3 model mice [32] and SCA1 PCs [33]. At the same time maintenance of the DSE obviously depends on the contribution of BG which is predominantly compromised when the glia is targeted directly.

Finally, LTD in PF-PC synapses was investigated. LTD is a complex process that requires low-frequency stimulation of PFs combined with strong postsynaptic depolarization. mGluR-mediated signaling in PCs is essential for LTD induction and it is impaired in B05 mice where the expression of mutant Ataxin-1 is limited to PCs [6]. In the present work, the impairment of LTD was demonstrated in the SCA1 KI model, where the expression is ubiquitous and, more intriguingly, in the LVV model where ATXN1[Q85] is selectively expressed in BG (Figure 7). We hypothesize that induction and maintenance of LTD strongly depend on mechanisms residing in BG (for example the activity of transporters, supply of glutamine for the synthesis of glutamate by presynaptic elements, etc). Therefore, LTD once again demonstrates the importance of BG in the pathogenesis of SCA1.

In summary, among all examined SCA1 models, the impairments found in SCA1 KI mice are the most prominent. It needs to be acknowledged that these animals carry the longest mutant Ataxin-1 (Q154). Our results point to specific mechanisms in SCA1 pathology that are not detectable in B05 mice. The present experiments demonstrate that 12-week-old SCA1 KI mice exhibit impaired short and long-term synaptic plasticity due to dysfunction of both neuronal and glial mechanisms. We speculate that reactive BG releases high quantities of S100β which can be neurotoxic [34,35,36]. Compromised BG may also fail to provide adequate K^+^ buffering, supply of glutamine, and inadequate extracellular glutamate removal. These factors lead to excitotoxicity.

## 5. Conclusions

In this work, we uncover new pathophysiological mechanisms of SCA1 that affect the PC membrane properties and plasticity in their synapses and reveal the contributions of BG to the SCA1 pathology. A better understanding of the mechanisms of SCA1-induced degeneration should help to develop new therapeutic approaches and increase the lifespan of SCA1 patients.

## Figures and Tables

**Figure 1 cells-11-02247-f001:**
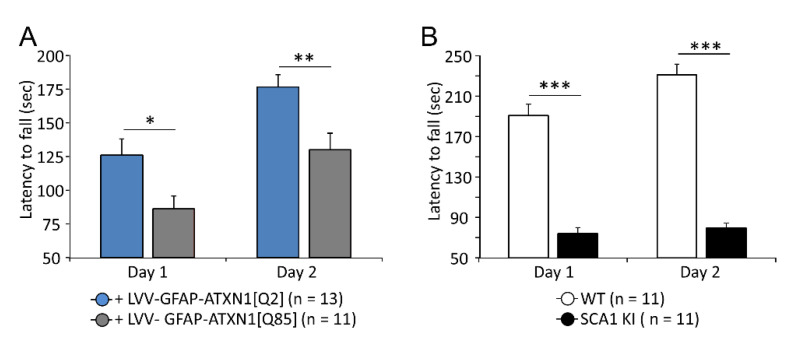
Rotarod performance in two SCA1 models. (**A**) Selective expression of ATXN1[Q85] in BG affected performance of mice on day 1 and 2 but did not prevent the improvement on day 2. (**B**) SCA1 KI mice performed poorly on day 1 and did not improve on day 2. The number of mice tested is shown in parentheses. Asterisks indicate statistically significant difference compared to controls, calculated using two-way ANOVA; * *p* < 0.05; ** *p* < 0.01; *** *p* < 0.001.

**Figure 2 cells-11-02247-f002:**
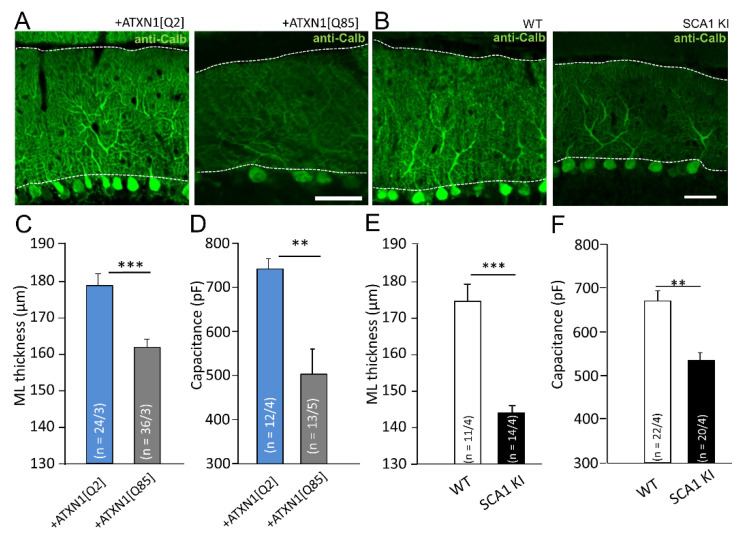
Mutant Ataxin-1 induces changes in PC morphology in SCA1 models. (**A**,**B**) Confocal images of cerebellar cortex labeled with anti-Calbindin (anti-Calb, PCs marker) show collapse of the molecular layer (ML) in mice that express ATXN1[Q85] selectively in BG (**A**) and in SCA1 KI mice (**B**). ML is marked by dotted lines. Scale bar 50μm. Average (**C**) depth of ML and (**D**) capacitance of PCs are reduced in mice that express ATXN1[Q85] selectively in BG and in SCA1 KI mice (**E**,**F**). The number (n) of tested areas or PCs and animals (areas/animals) or (PCs/animals) are shown in parentheses. Asterisks indicate a statistically significant difference compared to respective controls (*t*-test; ** *p* < 0.01; *** *p* < 0.001).

**Figure 3 cells-11-02247-f003:**
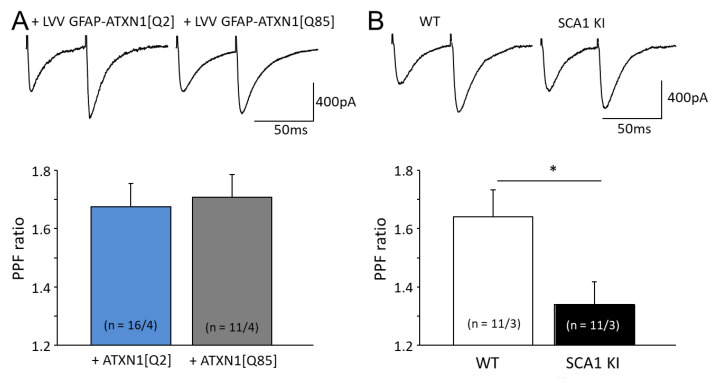
Impairment of presynaptic short-term plasticity in PF-PC synapses in SCA1 models. (**A**) Top—PPF in PC in response to PF stimulation, averaged traces. Below—PPF ratio is not affected in mice that express ATXN1[Q85] selectively in BG. (**B**) PPF is reduced in PF-PC synapses of SCA1 KI mice. The numbers (*n*) of tested PCs and animals (PCs/animals) are shown in parentheses. Asterisks indicate a statistically significant difference compared to respective controls (*t*-test; * *p* < 0.05).

**Figure 4 cells-11-02247-f004:**
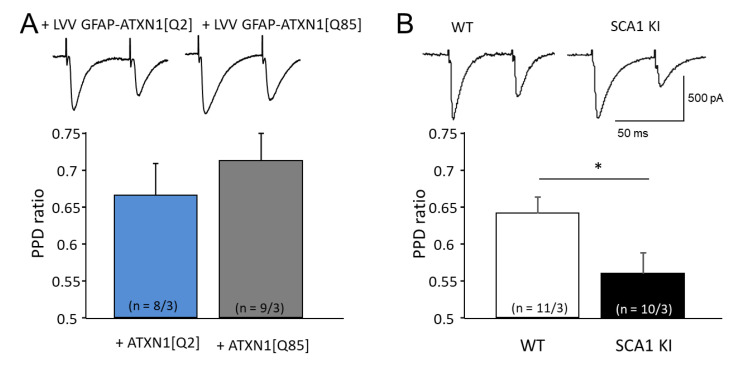
Impairment of presynaptic short-term plasticity in CF-PC synapses in SCA1 models. (**A**) PPD ratio is preserved in LVV-GFAP-Flag-ATXN1[Q85] injected mice as compared to LVV-GFAP-Flag-ATXN1[Q2] controls, indicating relatively preserved function of CF-PC synapses. (**B**) Enhanced PPD reflecting impaired presynaptic function of CF-PC synapses in SCA1 KI mice. The numbers (*n*) of tested PCs and animals (PCs/animals) are shown in parentheses. Asterisks indicate a statistically significant difference compared with respective controls (*t*-test; * *p* < 0.05).

**Figure 5 cells-11-02247-f005:**
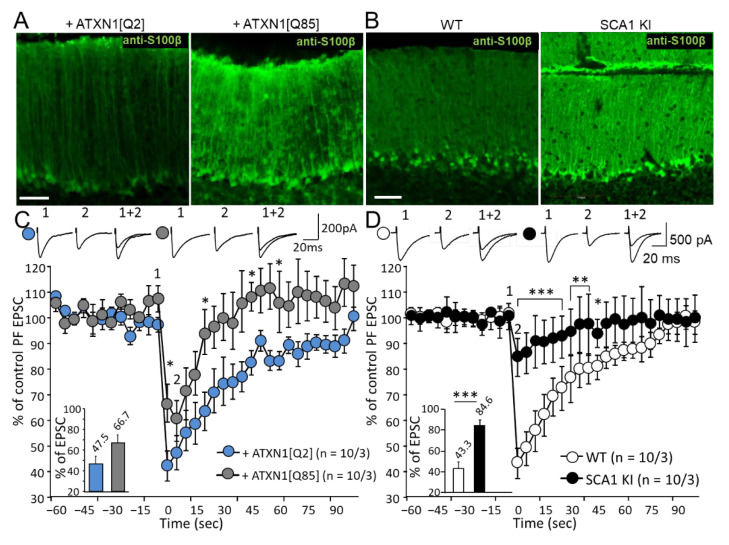
Reactive glia in SCA1 model mice alters depolarization-induced depression of excitation (DSE). (**A**,**B**) Confocal images of cerebellar cortex show the enhanced S100β expression in mice that express ATXN1[Q85] selectively in BG (**A**) and in SCA1 KI mice (**B**) Scale bar 50μm. (**C**,**D**) Average PF EPSC amplitudes before (1) and immediately after (2) depolarization. The amplitudes of PF EPSC were normalized to values before depolarization. Inserts show the average normalized amplitudes at time point 2. The number (*n*) of tested PCs and animals (PCs/animals) are indicated on the graphs. Asterisks indicate a statistically significant difference compared with controls (one-way ANOVA followed by Tukey’s post hoc test; * *p* < 0.05; ** *p* < 0.01; *** *p* < 0.001).

**Figure 6 cells-11-02247-f006:**
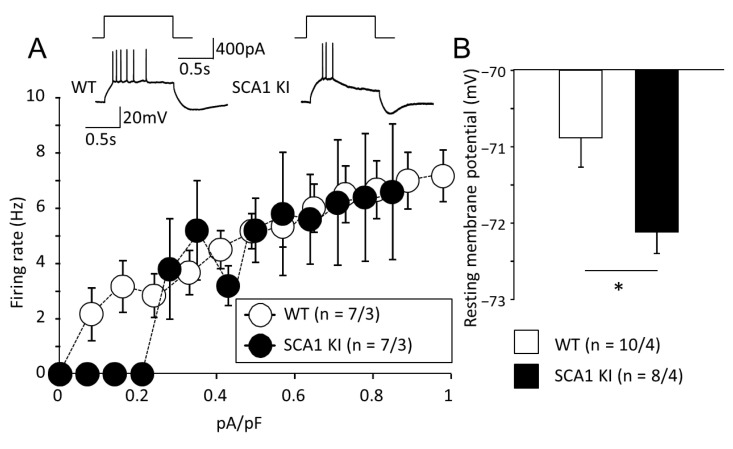
PCs from SCA1 KI mice are less excitable and slightly hyperpolarized. (**A**) The firing frequencies (in Hz) are plotted against the injected currents corrected for the membrane capacitance (pA/pF). Representative traces after the same current injection are shown above the graph. (**B**) Resting membrane potentials of PCs. The numbers (*n*) of tested PCs and animals (PCs/animals) are shown in parentheses. Asterisks indicate a statistically significant difference compared with WT control mice (*t*-test; * *p* < 0.05).

**Figure 7 cells-11-02247-f007:**
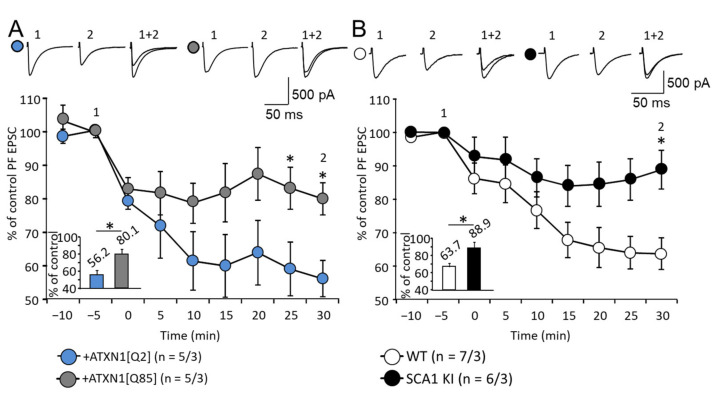
Impairment of LTD at PF-PC synapses in both SCA1 models. (**A**,**B**) Average time course diagram of PF EPSC amplitudes before and after LTD induction in mice that express ATXN1[Q85] selectively in BG (**A**) and in SCA1 KI mice (**B**). The amplitudes of PF EPSC were normalized to values before the LTP induction. The numbers (*n*) of tested PCs and animals (PCs/animals) are indicated in the graph. Representative PF EPSC traces are shown above the diagram. Time points: before (1) and 30 min after (2) LTD induction. PF EPSC averaged amplitudes 30 min after LTD induction are shown in the inserts. In both models, LTD was reduced. Asterisk indicates a statistically significant difference compared with respective controls (*t*-test; * *p* < 0.05).

**Table 1 cells-11-02247-t001:** Passive electrophysiological properties of PCs from 12-week-old mice. Cm—membrane resistance, Ra—access resistance, Rm—membrane resistance. *t*-test indicates significant effects in Cm. *n*—number of cells/number of animals, ** *p* < 0.001; ^†††^
*p* < 0.001.

Genotype	Cm (pF)	Ra (mΩ)	Rm (mΩ)
WT (*n* = 20/8)	670.4 ± 26.7	11.6 ± 0.6	239.4 ± 25.0
SCA1 KI (*n* = 22/8)	534.1 ± 27.6 **	10.7 ± 0.8	253.0 ± 30.7
+LVV-GFAP-Flag-ATXN1[Q2] (*n* = 12/3)	748.4 ± 22.6	11.1 ± 0.5	236.6 ± 13.2
+LVV-GFAP-Flag-ATXN1[Q85] (*n* = 13/3)	508.6 ± 57.5 ^†††^	11.4 ± 0.8	259.7 ± 18.0

**Table 2 cells-11-02247-t002:** AMPA receptor-mediated EPSCs in PC of 12-week-old mice. Kinetics of PF EPSCs and CF EPSCs (10–90% rise time and decay time constant) and their amplitudes are shown. *t*-test did not show any significant difference. *n*—number of cells/number of animals, *p* > 0.05.

Genotype	PF EPSC (pA)	Rise Time (ms)	Decay Time (ms^−1^)	CF EPSC (pA)	Rise Time (ms)	Decay Time (ms^−1^)
WT (*n* = 13/3)	844.8 ± 95.5	2.7 ± 0.2	10.3 ± 0.9	855.0 ± 67.7	1.3 ± 0.1	8.2 ± 0.5
SCA1 KI (*n* = 15/3)	743.2 ± 99.3	2.5 ± 0.4	12.2 ± 1.3	877.7 ± 110.1	1.2 ± 0.1	8.6 ± 0.3
+LVV-GFAP-Flag-ATXN1[Q2] (*n* = 9/3)	714.6 ± 84.1	2.4 ± 0.2	10.7 ± 2.0	767.9 ± 112.1	1.3 ± 0.3	11.4 ± 1.1
+LVV-GFAP-Flag-ATXN1[Q85] (*n* = 8/3)	801.9 ± 94.8	2.3 ± 0.2	13.6 ± 1.2	676.0 ± 86.5	1.4 ± 0.2	10.1 ± 1.0

## Data Availability

Not applicable.

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
