# Peer review of "Indirect Negative Effect of Mutant Ataxin-1 on Short- and Long-Term Synaptic Plasticity in Mouse Models of Spinocerebellar Ataxia Type 1"

_cells, 2022, doi:10.3390/cells11142247_

Round 1
Reviewer 1 Report
This article examines Mutant Ataxin-1 in Knock-In SCA Type I in mice. Overall, the article question examined is important and adequately addressed the methods. Statistical analysis is appropriate. Literature review is adequate.
There are a few minor revisions suggested:
Please review inconsistent use of rotarod and rota-rod. Change to rotarod throughout in manuscript and in figures, which is the more standard use.
Figure 1 should be a bar graph instead of a line graph given there is only 2 points examined.
The authors need to state gender distribution of samples as gender is known to play a role particularly in rotarod outcome. Also, when comparing results to literature the authors need to consider that not all researchers use accelerating rotarod - some use constant speed rotarod, which does impact results. (For example, see analysis from ALS mice - Pfohl 2015 - https://pubmed.ncbi.nlm.nih.gov/26594635/)
The abstract has too much background. It should be written to highlight each key quantitative result (key electrophysiology, key immunohistochemistry, key rotarod, etc.) ; also, the formatting of the abstract needs addressed with various changes in tone and use of spacing.
Need to cut back on first person use of “we” and write more consistently in third person. Occasional use of “we” is fine, but this is too much as currently written.
Optional: A small comparison of these results to ALS may be warranted in the Discussion. The electrophysiology results looks very similar to recent age-matched ALS transgenic mice studies.
Author Response
This article examines Mutant Ataxin-1 in Knock-In SCA Type I in mice. Overall, the article question examined is important and adequately addressed the methods. Statistical analysis is appropriate. Literature review is adequate.
There are a few minor revisions suggested:
Q: Please review inconsistent use of rotarod and rota-rod. Change to rotarod throughout in manuscript and in figures, which is the more standard use.
A: Thank you very much for the suggestion. We changed rota-rod to rotarod.
Q: Figure 1 should be a bar graph instead of a line graph given there is only 2 points examined.
A: We changed the figure 1 and put the bar graphs.
Q: The authors need to state gender distribution of samples as gender is known to play a role particularly in rotarod outcome. Also, when comparing results to literature the authors need to consider that not all researchers use accelerating rotarod - some use constant speed rotarod, which does impact results. (For example, see analysis from ALS mice - Pfohl 2015 - https://pubmed.ncbi.nlm.nih.gov/26594635/)
A: Thank you very much for the comments. All groups of mice contained males and females at approximately even numbers, no differences were noted and data was pooled. We wrote it in the Materials and Methods. In methodological works about rotarod you can find the advantages and disadvantages of accelerated mode. We used this mode because SCA1 KI mice have very prominent motor impairment and they more adapted to the slow speed during initiation of training. The disadvantage of a set speed mode is that some animals with poor coordination will fall off at the start, whereas for those that do stay on, the test will soon start to measure endurance rather than coordination per se (Deacon et al., 2013). We mentioned about it in Materials and Methods.
Q: The abstract has too much background. It should be written to highlight each key quantitative result (key electrophysiology, key immunohistochemistry, key rotarod, etc.); also, the formatting of the abstract needs addressed with various changes in tone and use of spacing.
A: Thank you very much for the advice. We rewrote the abstract according your comments.
Q: Need to cut back on first person use of “we” and write more consistently in third person. Occasional use of “we” is fine, but this is too much as currently written.
A: We excluded “we” and corrected the manuscript.
Q: Optional: A small comparison of these results to ALS may be warranted in the Discussion. The electrophysiology results looks very similar to recent age-matched ALS transgenic mice studies.
A: Indeed, there are many pathological interconnections between all neurodegenerative diseases. We found the “hot points” and cited papers about ALS in the manuscript.
Reviewer 2 Report
The manuscript: “Indirect negative effect of mutant Ataxin-1 on short- and long-term synaptic plasticity in mouse knock-in model of spinocerebellar ataxia type 1” describes a series of experiments aimed to reveal the new aspects of pathophysiology in SCA1 covering the synaptic properties and plasticity.
The manuscript is well written and understandable however I would like to address a few minor remarks:
- The lack of description how many animals was finally used for experiments in Materials and Methods section
- I would expect to know why the authors chosen this relative early period of postnatal mice development and maybe it is worth to highlight it in the manuscript title
- I think that it is woth to address and cite the article: Chandrakanth Reddy Edamakanti, … , Marco Martina, Puneet Opal: Mutant ataxin1 disrupts cerebellar development in spinocerebellar ataxia type 1 J Clin Invest. 2018;128(6):2252-2265. https://doi.org/10.1172/JCI96765. covering similar earlier studies.
Author Response
The manuscript: “Indirect negative effect of mutant Ataxin-1 on short- and long-term synaptic plasticity in mouse knock-in model of spinocerebellar ataxia type 1” describes a series of experiments aimed to reveal the new aspects of pathophysiology in SCA1 covering the synaptic properties and plasticity.
The manuscript is well written and understandable however I would like to address a few minor remarks:
Q: The lack of description how many animals was finally used for experiments in Materials and Methods section.
A: We counted the mice number and wrote it in the Matherials and Methods.
Q: I would expect to know why the authors chosen this relative early period of postnatal mice development and maybe it is worth to highlight it in the manuscript title.
A: According to the first works about SCA1 KI mice (Watase et al., 2002) the alteration of motor performance in these mice starts from 5 weeks. We found the same disturbance in B05 mice. Moreover at 12 age of weeks the ataxia starts to be prominent in B05 mice. At this stage we succeeded to rescue the ataxic phenotype using Baclofen (Shuvaev et al., 2016). For this reason here we pay attention for pathology of SCA1 KI mice at 12 weeks of age. Moreover all data in manuscript have comparison with B05 mice at this age. We don’t call this stage early because there is a prominent PCs morphology alteration and apoptosis.
Q: I think that it is worth to address and cite the article: Chandrakanth Reddy Edamakanti, … , Marco Martina, Puneet Opal: Mutant ataxin1 disrupts cerebellar development in spinocerebellar ataxia type 1 J Clin Invest. 2018;128(6):2252-2265. https://doi.org/10.1172/JCI96765. covering similar earlier studies.
A: Thank you very much for the link! We found the interconnection with our findings and cited it in the text.